# Recycled Glass as a Substitute for Quartz Sand in Silicate Products

**DOI:** 10.3390/ma13051030

**Published:** 2020-02-25

**Authors:** Katarzyna Borek, Przemysław Czapik, Ryszard Dachowski

**Affiliations:** Civil Engineering and Architecture Department, Kielce University of Technology, Al. 1000-lecia PP 7, 25-314 Kielce, Poland; p.czapik@tu.kielce.pl (P.C.); tobrd@tu.kielce.pl (R.D.)

**Keywords:** silicate products, brick, sand, lime, glass, microstructure, tobermorite

## Abstract

In 2016, an average of 5.0 tons of waste per household was generated in the European Union (including waste glass). In the same year, 45.7% of the waste glass in the EU was recycled. The incorporation of recycled waste glass in building materials, i.e., concrete, cements, or ceramics, is very popular around the world because of the environmental problems and costs connected with their disposal and recycling. A less known solution, however, is using the waste glass in composite products, including sand-lime. The aim of this work was to assess the role of recycled container waste glass in a sand-lime mix. The waste was used as a substitute for the quartz sand. To verify the suitability of recycled glass for the production of sand-lime products, the physical and mechanical properties of sand-lime specimens were examined. Four series of specimens were made: 0%, 33%, 66%, and 100% of recycled waste glass (RG) as a sand (FA) replacement. The binder mass did not change (8%). The research results showed that ternary mixtures of lime, sand, and recycled waste glass had a higher compressive strength and lower density compared to the reference specimen. The sand-lime specimen containing 100% (RG) increased the compressive strength by 287% compared to that of the control specimen. The increase in the parameters was proportional to the amount of the replacement in the mixtures.

## 1. Introduction

Managing wastes and resources is one of the main topics of research developing in the scientific community. Minimizing the necessity for extracting raw materials and maximizing the material life by promoting the reuse and recycling is a program that European and global markets are working on and gradually implementing. The concept behind this program is to turn waste into a valuable resource by designing products that can be easily recovered and reused as a raw material for the same, or a similar, industry [1]. Actions like these are an effective way to avoid pollution, reduce waste emissions, and gradually increase the environmental performance [2].

In Europe, the recycling of glass is one of the most advanced. In some European countries, almost 85% of glass containers—bottles and jars—are made from recycled material. Unfortunately in Poland, both used containers and crushed glass are mostly sent to municipal or illegal landfills. Managing waste, despite many attempts and undertakings, has not been acceptably resolved to this day [3].

To reduce the load on landfills, some of the recycled waste glass is used as a replacement for aggregate in building materials, being an effective recycling alternative [4,5,6,7,8]. Many scientists used waste glass in concrete as a substitute for fine or coarse aggregate in order to improve its features [9,10,11]. Arezki [12] used glass sand as an alternative for quartz sand in ultra-high-performance concrete to select the optimal glass sand (GS) combination, not only based on the highest packing density, but also based on producing optimal concrete properties (workability and compressive strength). Mixing glass with Portland cement can, however, cause a decline in concrete strength due to the alkaline reaction, which depends on time [13]. However, the almost zero porosity and non-hygroscopic character of waste glass, as a substitute for natural sand, improves the properties of self-compacting cement mortar, such as increased workability and penetration resistance of chloride ions, reduced drying shrinkage, and improved compressive strength after being exposed to elevated temperatures [14,15,16]

Using RG in cement materials is not without its downsides. The disadvantage of glass with a smooth surface is the weak bond between RG and the cement paste. Consequently, a concrete or mortar specimen usually shows higher porosities [17,18]. An increased size and volume of voids in the transition zone reduce the mechanical properties, such as compressive and flexural strength [19,20,21].

Such common failures limit the wider use of RG in cement materials. To promote the use of RG in construction, and particularly in the building materials market, it is essential to seek alternative building materials that do not only include cement, but also recycled aggregate, and other materials, such as tires and other production waste. This is economically advantageous and is part of the Green Public Procurement (GPP) strategy [22]. For some EU countries, the use of construction materials with a percentage of recycled material is mandatory in the construction of public works [23].

Soda-lime glass is rich in silica and can potentially be used in materials that are rich in this raw material, such as in the external and internal walls of buildings. They are sand-lime products. They match the idea of sustainable development [24] and allow for creating a healthy living space for people without causing degradation of the natural environment. The raw materials that build the sand-lime products are quartz sand (over 90%), quenched lime (8%), and water. After forming, the material is subjected to a hydrothermal treatment, i.e., a temperature of around 180–200 °C and a water vapor pressure of about 16 bar. As a result of the reaction between the sand, lime, and water, crystalline phases are created in silicates.

The phase composition in the sand-lime products is a vital factor because it influences the performance of the compressive strength, porosity, and absorption capacity [25]. The products of synthesis include the C-S-H (Calcium Silicate Hydrate) phase, most often found in sand-lime products, as well as tobermorite and xonotlite. According to the literature [14], 11.3 A tobermorite includes a significant amount of water, but at temperatures above 300 °C, it decomposes to 9 A tobermorite. Xonotlite is similar to 11.3 A tobermorite in terms of structure, but it contains 5 times less water than tobermorite and is shaped at a temperature of ≈220 °C–380 °C.

The fundamental composition of silicate products is subject to modification. Fang et al. [26] successfully used a sand substitute in the form of low-SiO_2_-content copper tailing.

The research conducted for the purposes of this article covered the characteristics of sand-lime products in which the basic source of active silica sand (FA) was substitute with household recycled waste glass (RG) from grinding jars. The goal of the research was to obtain sand-lime bricks with similar or better features and optimize the production costs by reducing energy consumption and decreasing the autoclaving temperature.

## 2. Materials and Methods

### 2.1. Lime

Lime is one of the binders that act as plasticizers. In this case, the binder was derived from the Trzuskawica Production Plant (Nowiny, Poland). Declared values of the basic properties of lime are presented in the Table 1.

### 2.2. Sand

Quartz sand of natural origin was the raw material, which was the source of silica, and at the same time, acted as an aggregate. It was obtained from deposits exploited at the Silicates Production Plant in Ludynia (Poland). Granulation was tested in accordance with the PN-EN 933-1 [28] standard for control purposes (Figure 1 and Figure 2). The analysis of the glanurometric composition was carried out using the sieve method. Based on this, it was stated that fine quartz sand of natural origin with a maximum grain size of less than 0.5 mm would be used for the test. According to the Unified Soil Classification System, this kind of sand with less than 5% fines is designated with the SW symbol. The mean density was 2.63 g/cm^3^. The result indicates the average value of two measurements carried out in accordance with PN-EN-1097-6: 2013-11 [29].

### 2.3. Recycled Waste Glass

Transparent white recycled glass waste was obtained from the mechanical grinding of food storage jars. Transparent jars that were washed, dried, and crushed were received from households. The size of the glass particles was adjusted to the size of the quartz sand particles in the reference sand-lime product. The shape of the glass particles (Figure 3) was polyhedral and irregular. Based on the EDS analysis shown in Figure 4, the recycled waste glass had a high content of Si, Na, Mg, Al, Ca, K, S, and C. The mean density was 2.47 g/cm^3^ (mean value of six results, determined using a helium pycnometer, manufacturer: Quantachrome Instruments Headquarters, Boynton Beach, Florida). 

### 2.4. Preparing the Sand-Lime Specimens with the Addition of Waste Glass

Proper amounts of raw materials were weighed for particular series (Figure 5, Table 2). Highly reactive burnt lime was mixed with the fine aggregate, which included different mixtures of FA and RG (RG33%, RG66%, RG100%). Water was added to the mixed ingredients of the raw material in a quantity sufficient to put out the lime. The mixture was placed in a sealed glass vessel and dried in the dryer at 65 °C for 1 h. After the mixture reached the ambient temperature, water was added again in the amount necessary to obtain a mass with 6–8% moisture. Then, cylindrical specimens with a diameter and height of 25 mm were formed using two-stage and two-sided compression with inter-venting at a pressure of 10 MPa and 20 MPa. The specimens were autoclaved at 180 °C and at a saturated steam pressure of 1.002 MPa. Heating of the specimens at this temperature lasted 2.5 h, while proper hardening took 8 h. The specimens were taken from the autoclave after 12 h to cool the autoclave down and bring the samples to an ambient temperature of approx. 21 °C. 

### 2.5. Testing Methods 

The experimental specimens were tested for their physical and mechanical parameters. Mechanical tests were conducted in laboratory conditions at room temperature using a hydraulic press, model Controls 50-C9030 (manufacturer Controls, Warsaw, Poland). According to PN-EN 772-1: 2011 [30], compressive strength was tested 21 days after conducting the autoclaving process; the results are presented as the arithmetic mean of six measurements including the standard deviation. 

The water absorption was determined according to PN-EN 772-21:2011 [31], where the volume density was determined through hydrostatic method. The specific density was determined using a helium pycnometer (manufacturer Quantachrome Instruments Headquarters, Boynton Beach, Florida). The density results were determined using the average of four measurements. 

Morphological studies were conducted with a scanning electron microscopy (SEM-type Quanta 250 FEG, FEI, Hillsboro, Oregon, USA) using signals gathered by secondary electron (SE) and backscattered (BSE) detectors. The acceleration voltage was 5 kV. The pictures from the scanning areas were enriched by establishing the semi-quantitative chemical composition of the observed phases using X-ray energy dispersion spectroscopy (EDS, manufacturer FEI, Hillsboro, Oregon, USA). The measurements were conducted on flat cut surfaces. Samples were not covered with the metallic layer. 

The analytical X-ray diffractometry (XRD) method was used in order to identify the phases appearing in the tested specimens, using an Empyrean PANALYTICAL device (manufacturer Panalytical, Almelo, Netherlands) with a Cu lamp. 

## 3. Results

The following graphs (Figure 6, Figure 7 and Figure 8) provide the average values of the obtained test results, including the standard deviations. Figure 6 shows the change in compressive strength of the sand-lime products depending on the amount of RG in the specimen. The compressive strength increased with the amount of waste glass in the specimen. The strength for RG100% was found to be more than 7 MPa higher than the reference sample (R). A small increase in strength of 1.4 MPa was observed with RG33% in the specimen.

The volume density decreased with the increase of the RG content in the specimen (Figure 7). The R specimen had a volume density of 1.9 g/cm^3^. The volume density of RG100% was lower at 1.6 g/cm^3^. The addition of RG in silicates also reduced the specific density of the finished sand-lime products. The R specimen had a value of 2.6 g/cm^3^, while RG100% had a 15.7% lower specific density (2.2 g/cm^3^). Therefore, the difference in the density of the silicate samples tested was greater than the difference in aggregates used to make them (sand and white glass amounted to 6.1%). It can therefore be concluded that the density of glass was only one of the factors affecting the decrease in the specific and volume density. It can also be influenced by the construction of the contact zone between the binder and the aggregate, as well as the reaction of the aggregate used with the binder, which may affect the porosity of the silicate. Changes in the binder–aggregate contact zone have also been noticed in the work Powęzka and Szulej [32]. However, by considering the results of water absorption, it was found that the proportion of open pores increased considerably (Figure 8). The increase of RG content in the specimen reduced the amount of closed intra-grain pores. This was connected with the structure of the glass. Sharp-edged and smooth surfaces, however, made the formation of a compact structure more difficult. Expanding the knowledge on the porosity of finished sand-lime products with the inclusion of RG is the next stage of research.

Figure 9 compares the diffractograms obtained for silicates R, RG33%, RG66%, and RG100%. For all cases, except for the specimen without sand, the main obtained peaks were quartz and calcite. Portlandite was also discovered in all samples containing recycled waste glass. Its remnant suggests that during the autoclaving, not all the lime was able to react.

The intensity of the quartz peaks decreased along with the increase of the glass content in the specimen, until they disappear completely when 100% of the sand was replaced with glass. Simultaneously, as the amount of glass increased, the background diffractograms in the 2θ angle range = 15°–40° increased as well. This proved the replacement of crystalline quartz present in sand with the amorphous glass.

Figure 10 shows the BSE of the RG100% specimens. Cracked glass particles slowly reacting under hydrothermal conditions were visible. Light particles (1) were unreacted sodium glass grain cores with the main components of silicon, sodium, and oxygen (Figure 11). Darker glass particles (2), with increased calcium content, became separated from the core (Figure 12). The increased content of lime may have resulted from its attachment to the glass during the autoclaving process. Inside the cracks between the glass particles (3), lime-rich glass reacting with products in the presence of lime, were visible (Figure 13).

Figure 14 presents a picture of a RG33% microstructure. At the surface with RG particles (1), there was both a spongy amorphous C-S-H phase (2) and tobermorite crystals (3), similar to thin plates and at unevenly set needles. The tobermorite crystals had different shapes. The result indicates that adding RG did not slow down the formation of phases that are characteristic of silicate products.

## 4. Conclusions

Based on the above results, the following conclusions can be drawn:RG can be used as a substitute for quartz sand in sand-lime products. A noteworthy increase in compressive strength compared with the reference specimen indicated that it was possible to completely replace the quartz sand in the silicate mix with the recycled waste aggregate. The larger compressive strength was obtained while reducing the density. This fits the current trends in the development of building materials, which are based on both economic and construction considerations: less material consumption, easier transport, the possibility of building taller buildings, and decreasing the cross-section of structural elements.For the strength properties and density, the point of view for the silicates production, completely replacing the quartz sand with waste glass was the most advantageous. This operation allowed for obtaining a silicate with a 287% higher compressive strength with a decrease in density by more than 15% compared to a traditional silicate.The increase in the content of RG in sand-lime products caused a significant increase in water absorption.Research on the microstructure showed that the exchange of quartz sand for recycled waste glass did not influence the formation of the C-S-H phase and tobermorite in autoclaved silicate products.The properties of small silicate samples were analyzed in this research. To better understand the silicates’ properties in practice, further research will be carried out on larger volume samples. Comparing the impact of different types of glass on the sand-lime properties is also planned.

## Figures and Tables

**Figure 1 materials-13-01030-f001:**
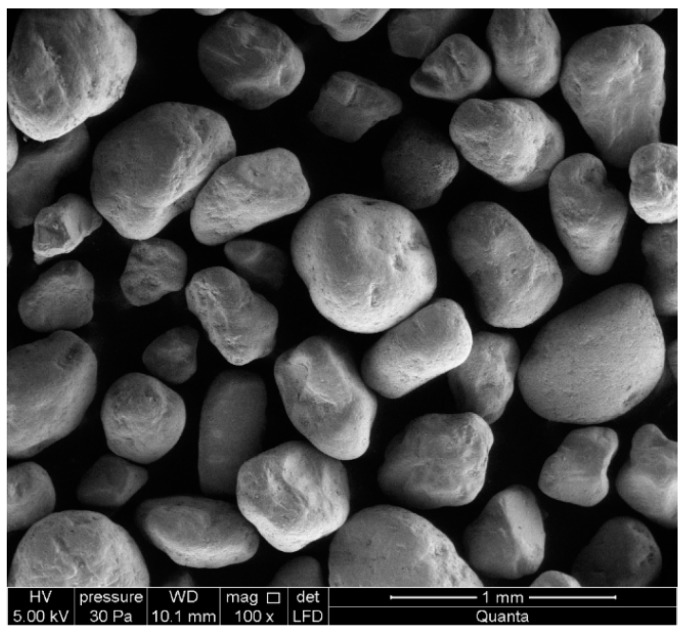
Grains of quartz sand.

**Figure 2 materials-13-01030-f002:**
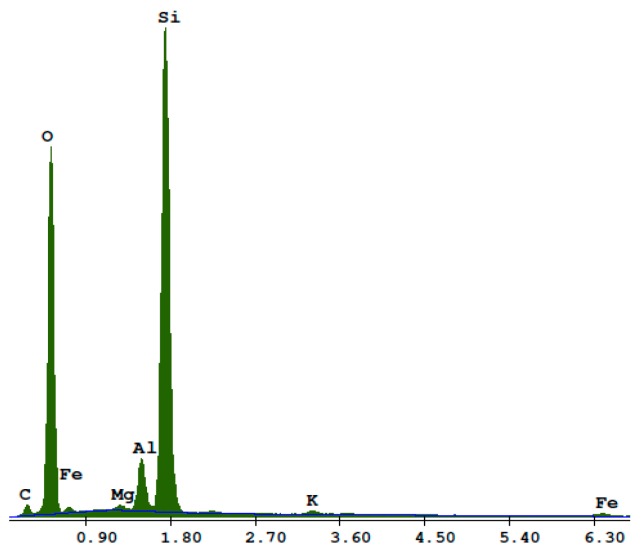
Energy dispersion spectroscopy (EDS) analysis of the quartz sand.

**Figure 3 materials-13-01030-f003:**
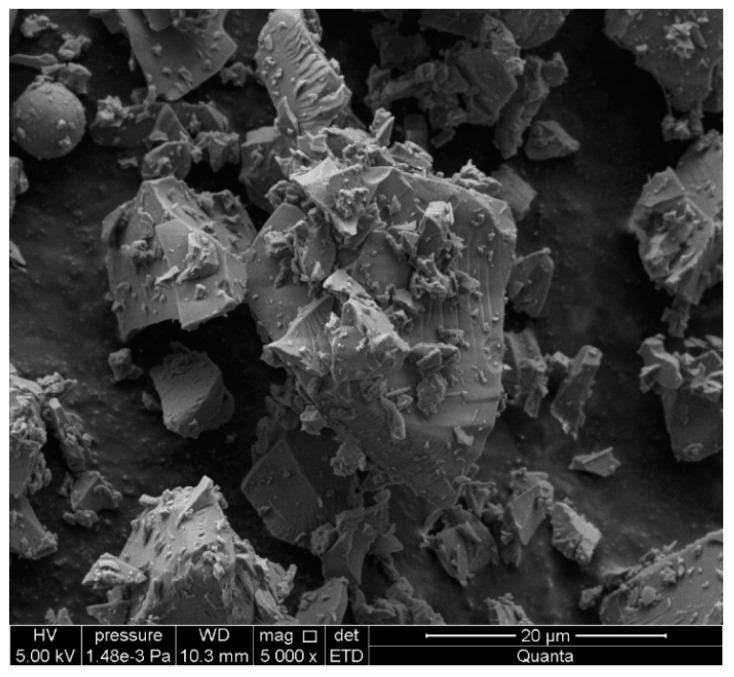
Recycled waste glass.

**Figure 4 materials-13-01030-f004:**
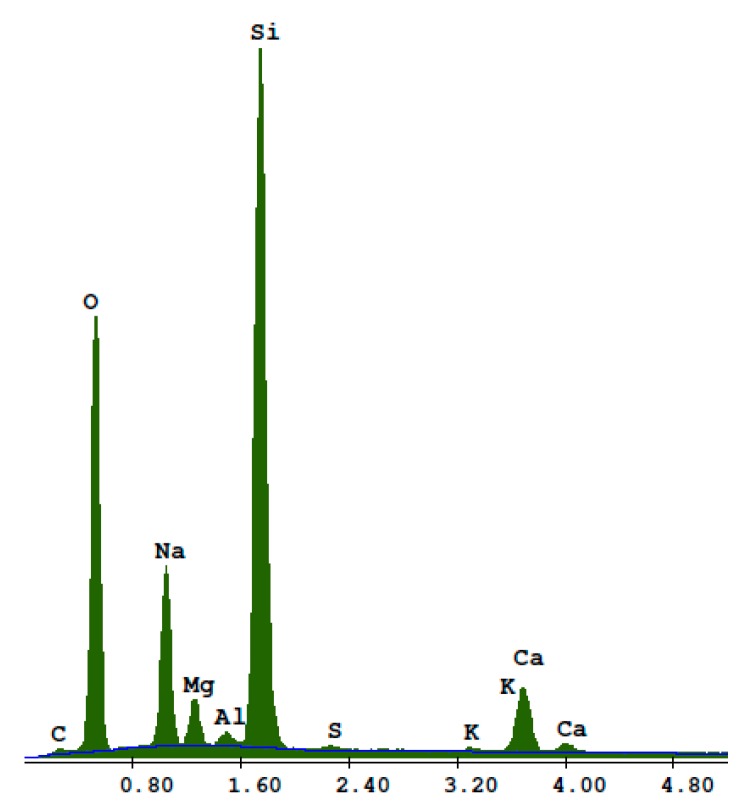
EDS analysis of the recycled waste glass particles.

**Figure 5 materials-13-01030-f005:**
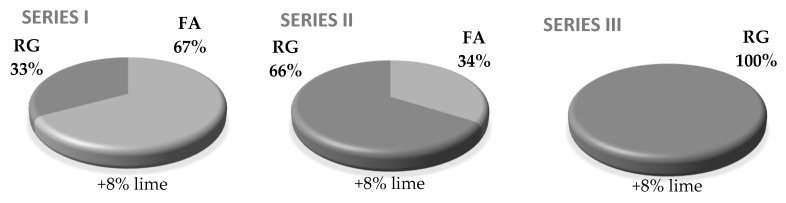
Quantitative summary of the raw material mix in particular series. FA: Sand, RG: Recycled Glass.

**Figure 6 materials-13-01030-f006:**
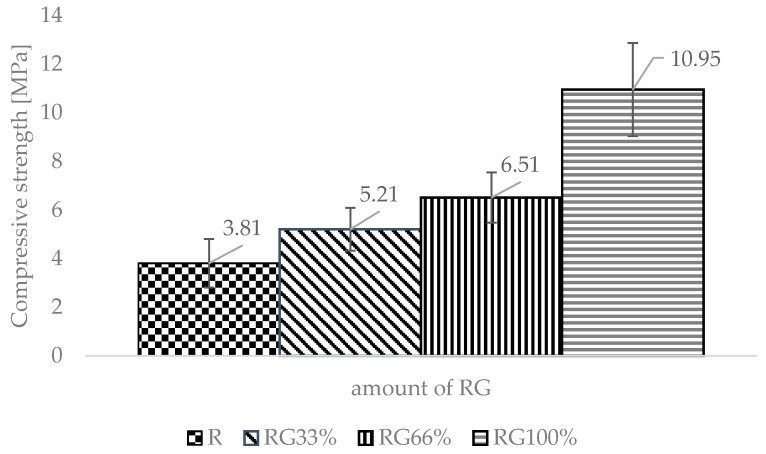
Compressive strength results.

**Figure 7 materials-13-01030-f007:**
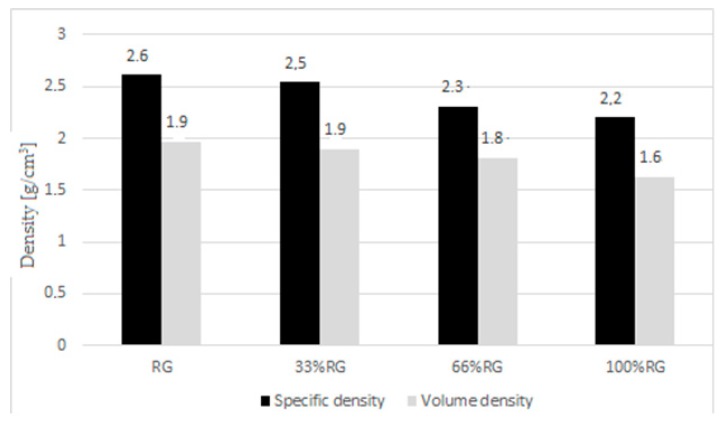
Density results.

**Figure 8 materials-13-01030-f008:**
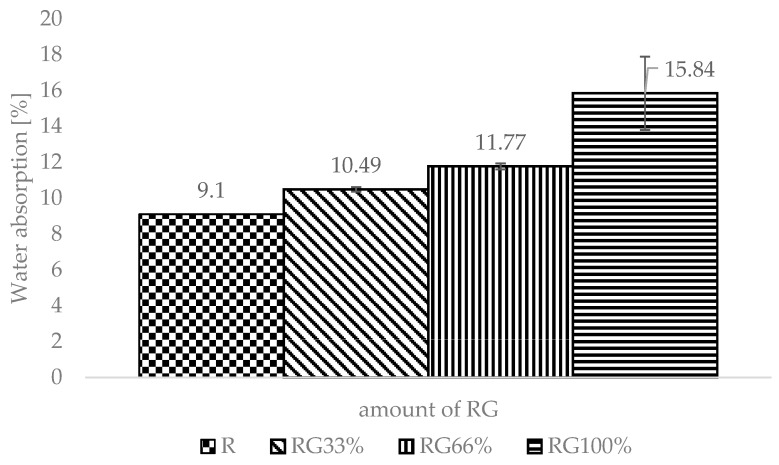
Water absorption results.

**Figure 9 materials-13-01030-f009:**
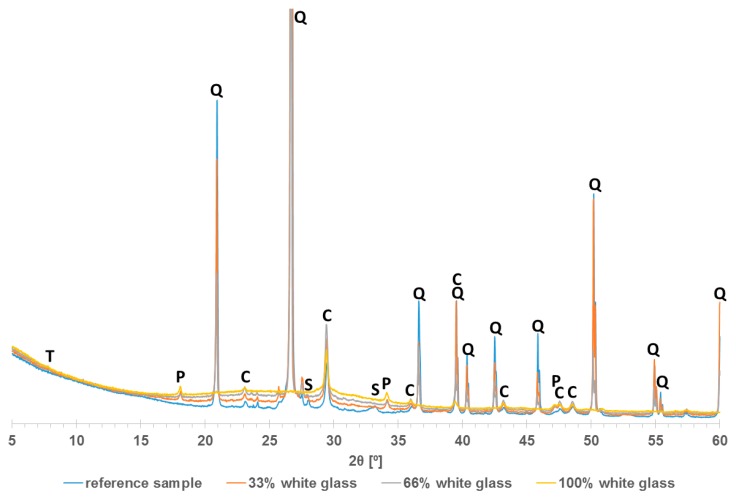
Diffractograms of silicate with different content of recycled waste glass. C—calcite, Q—quartz, P—portlandite, S—spurrite, T—tobermorite.

**Figure 10 materials-13-01030-f010:**
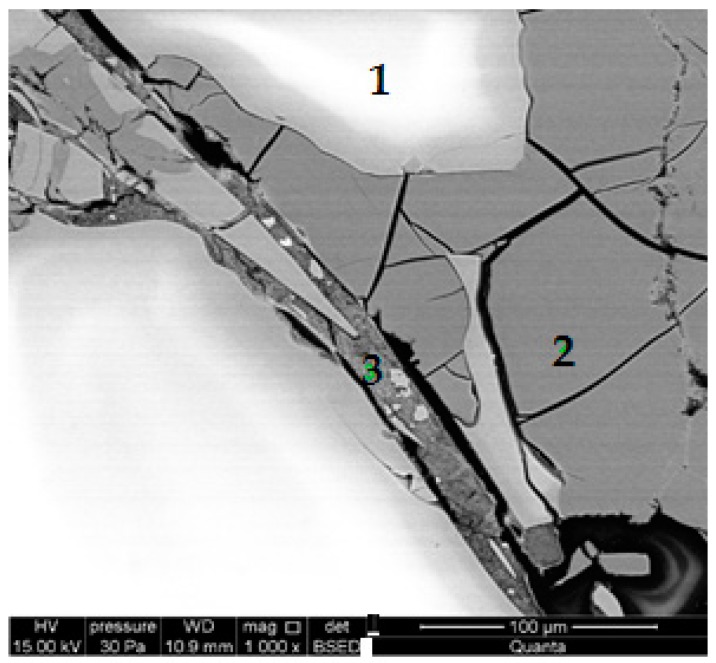
Backscattered electron (BSE) image of a RG100% specimen.

**Figure 11 materials-13-01030-f011:**
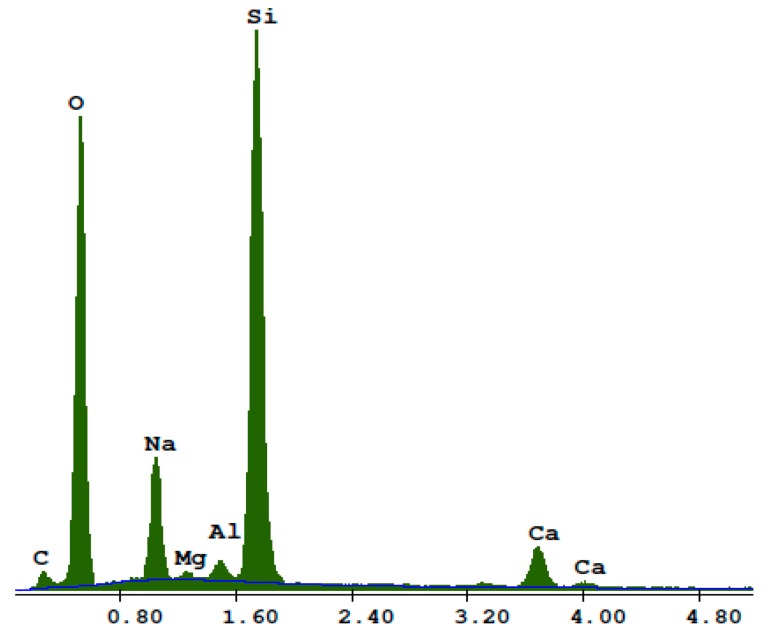
EDS analysis for point 1 in a RG100% sample.

**Figure 12 materials-13-01030-f012:**
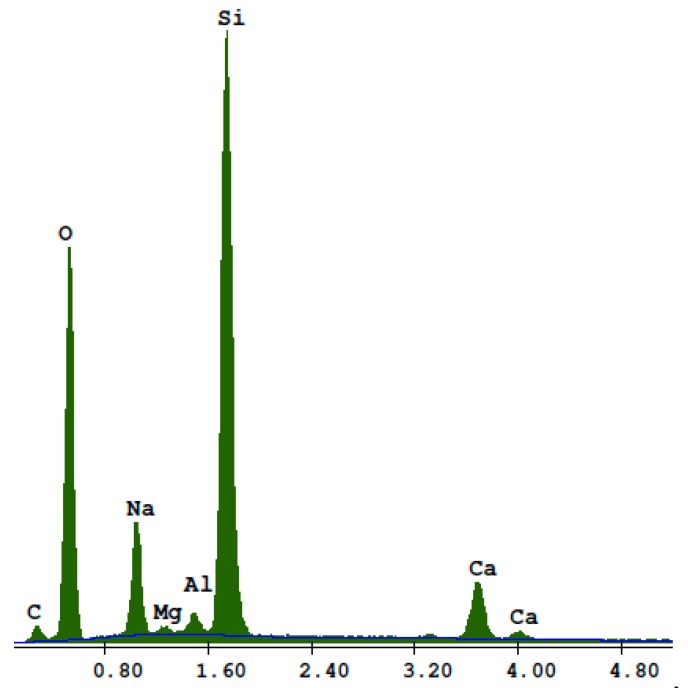
EDS analysis for point 2.

**Figure 13 materials-13-01030-f013:**
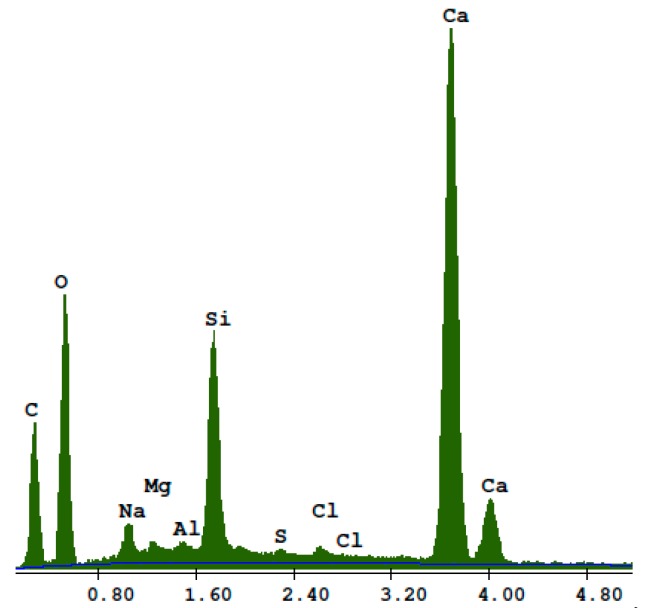
EDS analysis for point 3.

**Figure 14 materials-13-01030-f014:**
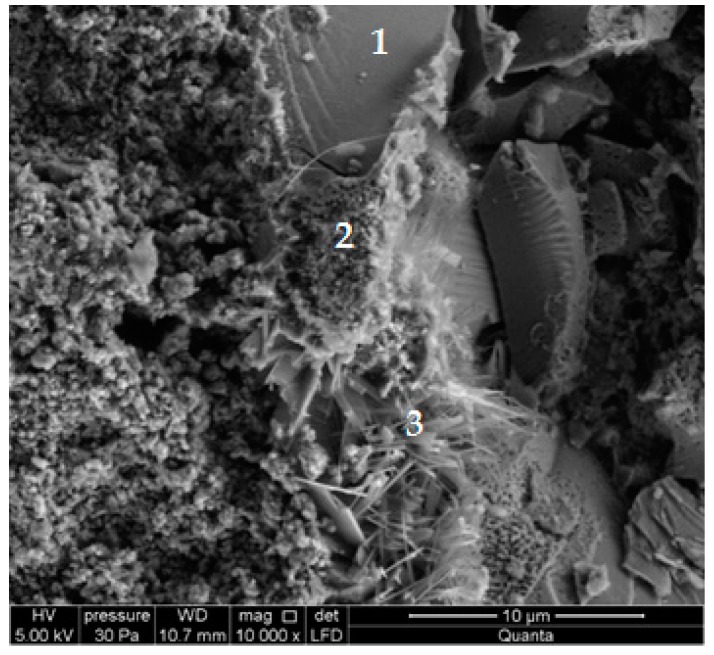
Microstructure of a RG33% specimen.

**Table 1 materials-13-01030-t001:** Summary of the basic properties of highly reactive burnt lime [27].

Functional Features of Burnt Lime	Declared Value
CaO + MgO (%)	≥91
MgO (%)	≤2.0
CO_2_ (%)	≤3.0
SO_3_ (%)	≤0.5
Screening through a 0.09 mm sieve (%)	≥90
Reactivity at 60 °C	≤2.0

**Table 2 materials-13-01030-t002:** Quantitative summary of FA and RG content, including the fractions.

Sieve Size	Series I RG33%		Series II RG66%		Series III RG100%	
	FA (g)	RG (g)	FA (g)	RG (g)	FA (g)	RG (g)
0	0.11	0.05	0.05	0.11	0	0.16
0.063	4.56	2.24	2.31	4.49	0	6.8
0.125	57.38	28.26	29.12	56.52	0	85.64
0.25	155.36	76.52	78.84	153.04	0	231.88
0.5	49.45	24.35	25.09	48.71	0	73.8
1.0	1.15	0.57	0.58	1.14	0	1.72

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
