# Peer review of "Recycled Glass as a Substitute for Quartz Sand in Silicate Products"

_materials, 2020, doi:10.3390/ma13051030_

Round 1

Reviewer 1 Report

The paper entitled “Recycled glass as a substitute for quartz sand in silicate products” investigates the suitability of recycled glass for the production of lime-sand products and examines the physical and mechanical features of such lime-sand specimen. The research results showed that ternary mixtures of lime, sand and recycled waste glass have higher compressive strength and lower density compared to the reference specimen. The optimal value of the most important examined trait i.e. compressive strength, with (RG) equal to 100% in the sand-lime raw material, which increases this value over 2.5 times comparing to the control specimen, is 10.95 MPa. The increase in the parameters is proportional to the amount of the extract in the mixtures.

The paper is interesting.

Comments:

Some editing of the English language and style is required.

There are two (!) figures numbered as 12:

Figure 12. EDS analysis for point 2. (Page 8).

Figure 12. Microstructure of a RG33% specimen. (Page 9).

The second one (on page 9) should be numbered as Figure 14.

Author Response

Detailed Response to Reviewer 1:

Thank you for your constructive comments on our manuscript.

We have addressed your comments one by one below and revised our paper based on your suggestions. In the article the revisions are shown in the yellow colour.

We hope that our revisions will be satisfactory:

  1. Some editing of the English language and style is required.

Thank you for your comment. English language has been corrected. The article was submitted for correction.

  1. There are two (!) figures numbered as 12:

Figure 12. EDS analysis for point 2. (Page 8).

EDS analysis for point 2

Figure 12. Microstructure of a RG33% specimen. (Page 9).

The second one (on page 9) should be numbered as Figure 14.

Thank you for your comment. Sorry for our mistake. Description Figure 14 has been corrected.

Reviewer 2 Report

some minor revision to the text

line 10/11

"45.7% of the waste in the EU and 37.8%" (of...???) "were recycled.

line 17/18

"weew made: 33%, ...."

line 31/32

"resource, by..."

line 43

the author has to introduce the full term before using abbreviation GS

line 56/58

the concept is presented in a reductive way. the topic needs further consideration in relation to the growing attention to the circular economy, GPP, GPP criteria toolkit. (i.e. Uttam et al, 2014, "Green public procurement (GPP) of construction and building materials")

For some EU countries, the use of construction materials with % of recycled material is mandatory in the construction of public works (i.e. Manzone et al., 2019, "The Italian Response to Sustainability in Built Environment: The Match between Law and Technical Assessment" ) ... for this reason the producers' market is slowly moving in this direction (line 57/58: "to seek alternative to building material that do not include cement")

statement of line 57/58 cuold be better explained: i.e. "that do not includes only cement, but also recycled aggregate, and other materials as tyres and other production waste..."

line 123: for a better view, separate the table from the graphs

line 142: chapter 3???

method and tests are presented in a simple way; adding a few phrases (maybe obvious for experts in the field of materials), the article can also be  read easily by those who are generally interested in the market of new building products using recycled materials.

line 197: chapter 4???

too short. conclusions can explain further may contain future research findings, e.g. costs/opportunities for use, convenience of building material manufacturers, levels of use in construction, ...

Author Response

Detailed Response to Reviewer 2:

Thank you for your constructive comments on our manuscript.

We have addressed your comments one by one below and revised our paper based on your suggestions. In the article the comments are shown in the yellow colour.

We hope that our comments will be satisfactory:

  1. line 10/11

"45.7% of the waste in the EU and 37.8%" (of...???) "were recycled.

Thank you for your comment. In the same year, 45.7% of the waste glass in the EU were recycled.

  1. line 17/18

"weew made: 33%, ...."

Thank you for your comment. It has been corrected.

  1. line 31/32

"resource, by..."

Thank you for your comment. It has been corrected: “resource by…”

  1. line 43

the author has to introduce the full term before using abbreviation GS

Thank you for your comment. The abbreviation has been explained. (GS) means glass sand.

  1. line 56/58

the concept is presented in a reductive way. the topic needs further consideration in relation to the growing attention to the circular economy, GPP, GPP criteria toolkit. (i.e. Uttam et al, 2014, "Green public procurement (GPP) of construction and building materials")

For some EU countries, the use of construction materials with % of recycled material is mandatory in the construction of public works (i.e. Manzone et al., 2019, "The Italian Response to Sustainability in Built Environment: The Match between Law and Technical Assessment" ) ... for this reason the producers' market is slowly moving in this direction (line 57/58: "to seek alternative to building material that do not include cement")

statement of line 57/58 cuold be better explained: i.e. "that do not includes only cement, but also recycled aggregate, and other materials as tyres and other production waste..."

Thank you for your notes. Comments have been introduced on lines 58-61. References have been added to the article.

  1. line 123: for a better view, separate the table from the graphs

Thank you for your comment. The table has been presented in full.

  1. line 142: chapter 3???

Thank you for your comment. Chapter has been changed.

  1. method and tests are presented in a simple way; adding a few phrases (maybe obvious for experts in the field of materials), the article can also be read easily by those who are generally interested in the market of new building products using recycled materials.

Thank you for your comment. Relevant literature is provided for each method. Unfortunately, X-ray and microstructural analysis is a complex issue. It is difficult to explain their methodology in a few words. We assume that the readers have a basic knowledge about field of building, know the literature and understand the methodology.

  1. line 197: chapter 4???

Thank you for your comment. Chapter has been changed.

  1. too short. conclusions can explain further may contain future research findings, e.g. costs/opportunities for use, convenience of building material manufacturers, levels of use in construction, ...

Thank you for your comment. Conclusions have been developed. Please note in the article has been presented the laboratory research. Semi-industrial tests are planned at Silicate Production Plant. Economic conclusions will be drawn after them. The directions of our research are permanently consulted with the industry (Silicate Production Plants). The main importance in the analysis of wall products are multi-criteria assessments, which are planned for the future article.
